Canopy position has a profound effect on soybean seed composition

Huber Steven C. steven.huber@ars.usda.gov 1 2 3
Li Kunzhi 1 2 4
Nelson Randall 3 8
Ulanov Alexander 5
DeMuro Catherine M. 1
Baxter Ivan ivan.baxter@ars.usda.gov 6 7
1 Global Change and Photosynthesis Research Unit, United States Department of Agriculture, Agricultural Research Service , Urbana , IL , United States
2 Department of Plant Biology, University of Illinois at Urbana-Champaign , Urbana , IL , United States
3 Department of Crop Sciences, University of Illinois at Urbana-Champaign , Urbana , IL , United States
4 Lab of Plant Nutrition Genetic Engineering, Kunming University of Science and Technology , Kunming , Yunnan , China
5 Metabolomics Facility, Carver Biotechnology Center, University of Illinois at Urbana-Champaign , Urbana , IL , United States
6 Plant Genetics Research Unit, United States Department of Agriculture Agricultural Research Service , St. Louis , MO , United States
7 Donald Danforth Plant Science Center , Creve Coeur , MO , United States
8 Soybean/Maize Germplasm, Pathology, and Genetics Research Unit, United States Department of Agriculture, Agricultural Research Service , Urbana, IL , United States
Röder Marion
Electronic publication date: 2016 Sep 13
Publication date: 2016
Volume: 4
Electronic Location ID: e2452
Received 2016 Mar 30; Accepted 2016 Aug 16
Copyright: ©2016 Huber et al.
Copyright year: 2016
Copyright holder: Huber et al.
License: This is an open access article distributed under the terms of the Creative Commons Attribution License, which permits unrestricted use, distribution, reproduction and adaptation in any medium and for any purpose provided that it is properly attributed. For attribution, the original author(s), title, publication source (PeerJ) and either DOI or URL of the article must be cited.
License URL: https://creativecommons.org/licenses/by/4.0/

Keywords: Soybean, Canopy, Physiology, Elemental composition, Ionome, Nutrition

Funding: United Soybean Board and the US Department of Agriculture—Agricultural Research Service Funding was provided by the United Soybean Board and the US Department of Agriculture—Agricultural Research Service. The funders had no role in study design, data collection and analysis, decision to publish, or preparation of the manuscript.

==============================
Although soybean seeds appear homogeneous, their composition (protein, oil and mineral concentrations) can vary significantly with the canopy position where they were produced. In studies with 10 cultivars grown over a 3-yr period, we found that seeds produced at the top of the canopy have higher concentrations of protein but less oil and lower concentrations of minerals such as Mg, Fe, and Cu compared to seeds produced at the bottom of the canopy. Among cultivars, mean protein concentration (average of different positions) correlated positively with mean concentrations of S, Zn and Fe, but not other minerals. Therefore, on a whole plant basis, the uptake and allocation of S, Zn and Fe to seeds correlated with the production and allocation of reduced N to seed protein; however, the reduced N and correlated minerals (S, Zn and Fe) showed different patterns of allocation among node positions. For example, while mean concentrations of protein and Fe correlated positively, the two parameters correlated negatively in terms of variation with canopy position. Altering the microenvironment within the soybean canopy by removing neighboring plants at flowering increased protein concentration in particular at lower node positions and thus altered the node-position gradient in protein (and oil) without altering the distribution of Mg, Fe and Cu, suggesting different underlying control mechanisms. Metabolomic analysis of developing seeds at different positions in the canopy suggests that availability of free asparagine may be a positive determinant of storage protein accumulation in seeds and may explain the increased protein accumulation in seeds produced at the top of the canopy. Our results establish node-position variation in seed constituents and provide a new experimental system to identify genes controlling key aspects of seed composition. In addition, our results provide an unexpected and simple approach to link agronomic practices to improve human nutrition and health in developing countries because food products produced from seeds at the bottom of the canopy contained higher Fe concentrations than products from the top of the canopy. Therefore, using seeds produced in the lower canopy for production of iron-rich soy foods for human consumption could be important when plants are the major source of protein and human diets can be chronically deficient in Fe and other minerals.

Introduction

Although soybean seeds from a given plant may appear physically homogeneous, it has long been known that seed produced at the top of the canopy can have higher protein and less oil compared to seeds from the bottom of the canopy (Collins & Cartter, 1956). Subsequently it was demonstrated that positional effects are observed with determinate as well as indeterminate soybeans (Escalante & Wilcox, 1993a) and in normal protein as well as high-protein breeding lines (Escalante & Wilcox, 1993b). While these effects on protein and oil concentrations have been documented to occur, they are nonetheless not widely recognized today and there are no insights concerning possible physiological mechanisms that may underlie these positional effects. There are many other important seed constituents, in particular minerals, but the impact of canopy position on many of these seed constituents is unknown. Because legumes like soybean can contribute not only protein to the human diet but also minerals like iron (Fe) and zinc (Zn), canopy position effects on the concentrations of essential minerals could be important, especially for the health and nutrition of children and women. According to the World Health Organization, Fe deficiency is currently the most widespread mineral deficiency affecting more than 30% of the world’s population (http://www.who.int/nutrition/topics/ida/en/). One approach to control this problem is to increase Fe intake via dietary diversification with Fe-rich foods and it is possible that variation with canopy position could be exploited.

Several factors could affect the development of seeds at the top of the plant differently than those at the bottom of the canopy and therefore could be responsible for differences in seed composition at maturity. First, flowering in the indeterminate soybean plants as used in the present study occurs first at lower nodes; thus, there is the potential for seeds lower in the canopy to develop over a longer period. However, while there is a lot of information about node position and flowering, there are few reports that have documented differences in duration of the seed fill period (SFP) as a function of node, as was demonstrated in cultivar ‘Williams79’ (Raboy & Dickinson, 1987). A second factor is that seeds lower in the canopy also develop under altered environmental conditions in terms of temperature, irradiance, light quality and humidity, which are recognized to impact soybean seed composition (Carrera et al., 2009; Carrera et al., 2011; Wolf et al., 1982). A third factor is the contribution of remobilization of reserves, including minerals, from leaves that may vary among minerals and with node position. Therefore, the role of canopy microenvironment and node position on seed composition warrants further consideration.

In the present study, we grew a core group of ten soybean lines in Urbana, IL, over a 3-yr period and monitored seed composition (protein, oil and mineral element concentration) at maturity as a function of node position. In general, there was a continuum in composition with seed that developed at the top of the canopy having more protein but less oil and reduced concentrations of minerals such as Mg, Fe, and Cu compared to seeds produced at the bottom of the canopy. Of particular note was the variation in Fe concentration, which was generally ∼20% higher in seeds from the bottom of the canopy. The differences in mineral concentrations such as Fe could have direct impact on use of soybeans for human food in countries that primarily depend on plant protein sources for intake of minerals. We also tested several possible developmental and micro-environmental factors for their ability to influence the seed compositional gradients, and used metabolomic profiling of developing seeds to investigate biochemical determinants of the protein and oil gradients. Collectively, the results establish a new type of seed heteromorphism in soybean where seeds appear physically homogenous but differ in composition and provide new insights to some of the underlying factors that may be responsible for the gradients in composition from bottom to top of the canopy.

Materials and Methods

Plant growth and sampling

Soybean lines were grown at the University of Illinois South Farm, Urbana, IL, in a randomized complete block design with three replicates each year. Each plot consisted of three rows 2.5 m long, with 0.75 m between rows and a planting density of roughly 30 seeds m−1. To produce the thinning treatment, all but three plants were removed from each row shortly after flowering. Delaying thinning until after the reproductive period had begun minimized branching on the remaining plants. Approximately 20 cm of plants were thinned from the ends of each row and the third plant was left in the middle in the row. The remaining plants were spaced approximately 1 m apart.

Plants were harvested at maturity. All plants were cut close to ground level and brought into the laboratory. Each main stem was divided into four quadrants and the stem fractions in each quadrant were threshed together for each plot. Only normal-sized plants were included in the analysis, and extremely small, wrinkled or off-color seeds were manually removed from all samples before analysis.

Soy products

To produce flour, soybeans were blanched (boiled for ∼25 min) and then baked before grinding. To produce soymilk and okara (remaining solids), soybeans were blanched (boiled for ∼5 min) twice and then ground in water and cooled slightly. The soymilk (liquid phase) and okara (solid phase) were separated using a cheesecloth and then dried separately and reground before analysis.

Seed storage product analysis

Protein and oil were measured with an Infratech 1241 Grain Analyzer (FOSS Analytical AB, Höganäs, Sweden), which is a true Near Infrared Transmission instrument that generates a spectrum from 850 to 1,050 nm via the monochrome light source and mobile grating system. A 50-ml seed sample was used that allowed for 10 subsample readings reported on a 13% moisture basis.

Ionomic analysis

Seed analysis was conducted as described in Ziegler et al. (2013). Briefly, single seeds from each quadrant were weighed using a custom-built seed weighing robot and then digested in concentrated nitric acid before loading onto an Elan ICP-MS. Internal standards were used to control for differences in dilution and sample injection. Leaf and soy products were analyzed in the same manner except that samples were added to digestion tubes by hand and weighed. Custom scripts were used to correct for internal standards and correct for sample weight.

Metabolomic analysis

Metabolome analysis was done through Metabolomics Center, Roy J. Carver Biotechnology Center, University of Illinois at Urbana-Champaign. Frozen seeds of the cultivar ‘Williams 82’ with attached seed coats were homogenized in liquid nitrogen and about 25 mg FW was extracted at room temperature with 1 mL of 50% methanol followed by addition of 800 µ1 of methanol:chloroform (1:2) as outlined in File S7. Each extraction was followed by centrifugation (5 min at 15,000 g), and the supernatants were collected. With the exception of samples for analysis of coenzymes, final extracts were evaporated under vacuum at −60 °C and subjected to GC/MS analysis.

Metabolic profiling

Dried extracts were derivatized with 100 µL methoxyamine hydrochloride (40 mg ml−1 in pyridine) for 90 min at 50 °C, then with 100 µL MSTFA at 50 °C for 120 min, and following 2-h incubation at room temperature 5 µL of the internal standard (hentriacontanoic acid, 10 mg ml−1) was added to each sample prior to derivatization. Metabolites were analyzed using a GC-MS system (Agilent Inc, CA, USA) consisting of an Agilent 7890 gas chromatograph, an Agilent 5975 mass selective detector and a HP 7683B autosampler. Gas chromatography was performed on a ZB-5MS (60 m × 0.32 mm I.D. and 0.25 µm film thickness) capillary column (Phenomenex, CA, USA). The inlet and MS interface temperatures were 250 °C, and the ion source temperature was adjusted to 230 °C. An aliquot of 1 µL was injected with the split ratio of 10:1. The helium carrier gas was kept at a constant flow rate of 2 ml min−1. The temperature program was: 5-min isothermal heating at 70 °C, followed by an oven temperature increase of 5 °C min−1 to 310 °C and a final 10 min at 310 °C. The mass spectrometer was operated in positive electron impact mode (EI) at 69.9 eV ionization energy at m/z 30–800 scan range.

Amino acid analysis

A 20 µl aliquot of the internal standard DL-chlorophenylalanine (1 mg ml−1 in 0.1M HCI) was added to the extracts, dried under vacuum, derivatized with 50 µl of neat N-methyl and 50 µL of acetonitrile at 80 °C for 4 h, cooled to room temperature and centrifuged briefly to remove condensate from the top of tube prior to injection of 1 µL at 5:1 split ratio into the GC/MS system, which consisted of an Agilent 6890N (Agilent Inc, Palo Alto, CA, USA) gas chromatograph, an Agilent 5973 mass selective detector and Agilent 7683B autosampler. Gas chromatography was performed on a 60 m ZB-5MS column with 0.32 mm inner diameter (I.D.) and 0.25 µm film thickness (Phenomenex, CA, USA) with injection temperature and MSD transfer line of 230 °C both, and the ion source adjusted to 230 °C. The helium carrier gas was set at a constant flow rate of 2 ml min−1. The temperature program was 5 min at 150 °C, followed by an oven temperature ramp of 5 °C min−1 to 315 °C for a final 3 min. The mass spectrometer was operated in positive electron impact mode (EI) at 69.9 eV ionization energy in m/z 50–800 scan range. Acquired data were normalized to the internal standard (DL-p-chlorophenylalanine) and sample fresh weight. Amino acid concentrations were calculated based on 2–75 µg ml−1 standard curves.

Free fatty acids, total fatty acids and coenzymes were also measured and values obtained used in the global analysis, but specific results are not presented. Detailed methods for the analysis are available on request.

The spectra of all chromatogram peaks were compared with electron impact mass spectrum libraries NIST08 (NIST, MD, USA), W8N08 (Palisade Corporation, NY, USA), and a custom-built database (460 unique metabolites). All known artificial peaks were identified and removed. To allow comparison between samples, all data were normalized to the corresponding internal standard and the sample fresh weight (FW). The spectra of all chromatogram peaks were evaluated using the AMDIS 2.71 (NIST, MD, USA) program. Metabolite concentrations were reported as concentrations relative to the internal standard (i.e., target compound peak area divided by peak area of internal standard: NI = Xi × X−1IS) per gram sample weight. The instrument variability was within the standard acceptance limit (5%).

Metabolites with more than 50% of missing data were removed and for the rest of the metabolites, any missing data was imputed with one-half of the minimum positive value in the original data assuming their level was below the instrument detection limit. MVA and visualization was performed with SIMCA-P+ 12.0 software (Umetrics AB, Umeå, Sweden) and MetaboAnalyst (Xia & Wishart, 2011) using log-transformed and autoscaled data and validated by sevenfold Cross-Validation and permutation with 500 random. To address the problem of multiple comparisons the False Discovery Rate (FDR) test was adopted (Storey, 2002).

Data analysis

Protein, oil, and elemental data were analyzed using R and the packages dplyr, ggplot2, grid, reshape2, qtlcharts and gplots. All data and analysis scripts used in the analysis are included as a supplemental file and are available on www.ionomicshub.org.

Figure 1 Quadrants of a soybean plant.

The mature plant is divided up into quadrants upon harvest and each quadrant is analyzed separately. Plat normalized data uses the average of all four quadrants to normalize year, plot and line affects.

Figure 2 Canopy gradients of seed composition traits before normalization and line and year effects on total accumulation.

(A) Composition gradients from the bottom to the top of the canopy for cultivar ‘Chamberlain’. The plots display the quadrant average as a line with the 95% confidence interval calculated using standard error as the ribbon. Units are mg (Single seed weight), PPM (Fe) and percentage for Protein and Oil (B) Year and line effects for each compositional trait, represented as boxplots. Units are PPM.

Results

Canopy position affects soybean seed protein, oil and mineral concentrations

We investigated positional effects with a core group of ten soybean lines (Table S1) grown in Urbana, IL, over a 3-year period. Main stems were harvested at maturity and divided into four canopy position quadrants (Fig. 1) and the seeds collected from each quadrant were analyzed separately for major storage products (protein and oil) and various minerals. Representative results obtained for one cultivar (‘Chamberlain’) are presented in Fig. 2A with full plots provides as File S1. As shown, protein concentration increased with node position at which seeds developed going from bottom to top of the mainstem while oil and iron (Fe) concentration decreased. For both protein and oil, which are the major seed constituents, there was variation in the absolute concentrations among the 3 years of study, but general trends were similar. Differences in absolute concentrations among years were most apparent for protein concentration with highest levels obtained in 2010 and lowest in 2011, presumably reflecting the impact of weather on seed development and composition. Another confounding source of variation for canopy position analysis is genotype, and Fig. 2B highlights the substantial variation in absolute concentrations of seed constituents due to both genotype and year. As expected, absolute concentrations of Mg, S, K, P and Ca were highest (>1000 ppm); Mn, Fe, Rb, and Zn were intermediate (10 to 100 ppm), and Na, Co, Ni, Cu, Sr, Mo, and Cd were present at trace levels (<10 ppm).

In order to compare positional effects for various parameters across genotypes and years without the confounding effects of differences in absolute values, we normalized each canopy gradient to a mean value of one and the values for each quadrant were then expressed relative to the normalized mean. However, because the weather in each year of the study differed (Table S2), the normalized results for each parameter are presented separately for each year. Across the 10 soybean lines, oil concentration decreased progressively from bottom to top of the canopy and was associated with a reciprocal increase in protein concentration (Fig. 3A). Protein and oil concentrations in soybean seeds are usually inversely related (Wilcox, 1998) and this was apparent with variation within the canopy as well. Single seed weight (designated as sample weight in Fig. 3A) varied with canopy position with seed produced in the middle portion tending to be slightly heavier than seeds produced at either the bottom or top of the canopy; however, the storage product gradients were independent of seed weight variation. Storage product gradients did not vary significantly across the three years of the study; however, absolute protein and oil concentrations varied among the three years of the study (Fig. 2), This is perhaps a result of weather that differed substantially in terms of temperature and precipitation among the three growing seasons (Table S2).

Figure 3 Canopy gradients of seed composition traits.

For each trait, the data was normalized to the plot average to remove the effect of environment and genotype. The plots display the quadrant average as a line with the 95% confidence interval calculated using standard error as the ribbon. (A) Percentage protein, percentage oil and single seed weight. (B) Elements with a significant (p < 1e − 10) effect of gradient in an ANOVA analysis that included Entry, Year and Position.

We also found that canopy position significantly affected the seed ionome, which comprises all of the minerals and trace elements found in mature seeds (Fig. 3B and File S2). While there have been several studies of the soybean seed ionome (McGrath & Lobell, 2013; Myers et al., 2014; Sha et al., 2012; Ziegler et al., 2013), to our knowledge this is the first report demonstrating variation with canopy position. Figure 3B shows normalized canopy gradient plots for elements where there was a statistically significant (p < 0.01) variation in concentration with position. Several groups of minerals exhibited common responses with canopy position. The elements Mg, Fe, Cu, and Cd were present at highest concentrations in seeds from the bottom of the canopy and decreased progressively to the top of the canopy. Within this group, the profiles for Mg and Fe were similar to one another in that variation was relatively low and the gradients were almost identical across the three years; however, the relative changes in Fe concentration were much greater in magnitude compared to changes in Mg concentration. Cu, Zn and Cd showed similar patterns, but were more variable among years. The second group that was apparent included Ca and Sr, where seeds from the middle of the canopy exhibited the lowest concentrations except in 2010, when concentrations of both Ca and Sr tended to increase going up the canopy. Finally, Mn was alone in the third category that increased in concentration towards the top of the canopy in all 3 years. Ca and Sr, and Cd and Zn, are chemically similar which may explain their parallel profiles. It is interesting to note that while Rb is a chemical analog of K and the two are often closely correlated (Baxter, 2009), that was not the case for soybean seeds where significant position effects on Rb were observed (Fig. 3B) but not for K (see File S2). It is also noteworthy that 2010 was the one year where mineral profiles were often distinct from those in 2011 and 2012. All three years were above normal in terms of temperature, but 2010 was the only year with above normal precipitation. Thus, water availability may be a major environmental factor impacting positional effects on the seed ionome, and interestingly some minerals were affected (Ca, Mn, Cu, Zn, Sr) while others (Mg, Fe, Co, Rb, Cd) were not. We also measured other minerals (B, Na, Al, P, S, K, Ni, As, Se and Mo) that did not show statistically significant variation with nodal position and are presented in File S2.

Another way to compare canopy profiles for the minerals measured is to do an overall correlation matrix of quadrant variation normalized to plot averages. In this way, one can look across the entire data set for parameters that are correlated based on variation with nodal position. A strong positive correlation would indicate that both components changed not only in the same direction but also to the same relative extent. As shown in Fig. 4A, only a few strong correlations were apparent among the measured parameters. Variation in seed size (sample weight in Fig. 4A) did not significantly correlate with positional variation of any of the measured elements or storage products. Protein and oil concentrations were strongly negatively correlated, as expected. In terms of minerals and storage products, the quadrant variation in protein concentration correlated negatively with Fe and Cu, and positively with Mn, and the reciprocal pattern was apparent with oil concentration. Among the minerals, highly correlated element pairs included Fe–Cu, Ca–Sr, Ca–Mn, and Zn–Cu, and between P and S, Zn, and Co. As noted earlier, Ca and Sr are chemical analogs and frequently correlated (Baxter, 2009), but surprisingly, other chemical analog pairs such as K–Rb were not observed. Fe and Cu were positively paired and have been reported to be positively correlated in soybean seeds (Vasconcelos, Clemente & Grusak, 2014) but the basis for the pairing is unknown. Correlations between P and minerals are often considered to reflect association of the mineral with seed phytate, the principal form of P in seeds (Vreugdenhil et al., 2004).

Figure 4 Correlation plot among composition traits.

Pearson correlation values between compositional traits. (A) Correlation across 832 quadrants normalized to the plot average. (B) Correlation across 208 plot means.

In addition to comparing parameters based on quadrant variation, it is also worthwhile to compare plot averages, which will reflect genetic and environmental effects on absolute values of the parameters. Figure 4B shows a matrix plot of correlations between plot means. Compared to the corresponding plot that focused on quadrant variation (Fig. 4A), many more strong correlations were apparent when comparing plot means. For example, protein concentration was positively correlated with S and Zn (and more weakly with Fe). The correlation with S is expected as the total seed S has been shown to track closely with high cysteine- and methionine- containing proteins in the soybean seed (Krishnan et al., 2012). The correlations between protein content, Zn and Fe could be due to their primary role as cofactors of metalloproteins or to variation in senescence in leaves leading to nutrient remobilization (Uauy et al., 2006). Accordingly, there was a significant negative correlation of Fe, S, and Zn with oil concentration. Interestingly, there was also a strongly significant negative correlation of P with oil, whereas the positive correlation of P with protein concentration was relatively weak. The majority of mineral correlations were positive in nature, with a maxi-cluster of Rb, Mn, Sr, Mg, Ni, and Na and a mini-cluster of Fe, Cu and Zn. The mini-cluster pairs of Fe–Cu and Cu–Zn were noted in the plot of Fig. 4A, but several members of the maxi-cluster correlation were not reported in the plot normalized correlation matrix. For example, Mn and Mg concentrations did not relate to each other in terms of quadrant variation but were strongly positively correlated based on plot means, indicating that mineral uptake may be similar but allocation among seeds in different quadrants is controlled separately. Finally, P concentration exhibited a positive correlation with Mn, Fe, Cu, Zn, S and Co. The link among P and Zn, S and Co concentrations with quadrant variation was observed (Fig. 4A), but when analyzed in terms of plot means in Fig. 4B the association of P with Mn, Fe, and Cu became apparent as well. It is worth noting that in terms of plot means, there was no association between Ca and Sr suggesting that these chemical analogs do not always behave similarly. There was also a strong negative correlation between Mo and Sr, and Mo and S, perhaps suggesting a common component of the uptake system. Readers can explore all of the correlations and the underlying data in Files S3–S5.

Figure 5 Effect of thinning on compositional traits.

For each trait, the data was normalized to the plot average to remove the effect of environment and genotype. The plots display the quadrant average as a line with the 95% confidence interval calculated using standard error as the ribbon. (A) Percentage protein and percentage oil in 2010. (B) Elements (from 2010 to 2012) with a significant (p < 1e − 10) effect of gradient in an ANOVA analysis that included Entry, Year, Position and thinning.

Canopy microenvironment impacts seed composition

Our understanding of the environmental factors responsible for the positional effects on seed composition is limited; however, many microclimatic factors vary from the top to the bottom of the closed soybean canopy (Baldocchi, Verma & Rosenberg, 1983). Environment is well known to impact soybean seed protein and oil composition (Rotundo & Westgate, 2009). Therefore, we conducted experiments to broadly evaluate microclimatic differences within the canopy by thinning plants at flowering to remove the influence of neighboring plants. Removal of neighboring plants increased protein concentration at the expense of oil in seeds throughout the canopy of the spaced plants but the changes were greatest in pods lower on the main stem. As a result, the thinning treatment reduced the positional effect on protein and oil by 50–60% (Fig. 5A). Increased light energy to drive photosynthesis at most leaf positions and increased temperature at lower positions could both favor increased protein accumulation at lower nodes thereby reducing the difference between top and bottom seeds. However, while thinning significantly altered the main stem gradients in major storage products there was relatively little effect on minerals. As shown in Fig. 5B, the canopy positional effect on Mg, Fe and Cu was unaltered by the thinning treatment whereas Ca and Sr were similar to one another and showed a significant effect of thinning but only in one of the two test years (2010). The general conclusion is that thinning affects the canopy positional effect on some but not all minerals. This suggests that at least for Mg, Fe and Cu, the transport and homeostasis mechanisms are generally independent of instantaneous environmental factors and the transport of sucrose and amino acids into the developing seeds is not the sole factor driving their movement into seeds.

Seed fill period and seed composition

Another factor that could contribute to canopy position effects on seed composition is the duration of the seed-fill period (SFP), which is affected by genetic and environmental factors and is one of the major determinants of yield potential in soybean (Evans et al., 1995). Soybeans flower in response to photoperiod and the first flowers form lower in the canopy followed by flowering at upper nodes. Pods then form in the same order and when fully elongated the process of seed development is considered to begin when seeds are approximately 0.34 centimeter long (by visual inspection). In general, seeds lower in the canopy fill over a longer period but at a lower rate compared to seeds at the top of the canopy (Raboy & Dickinson, 1987) so that at maturity, final seed size tends to be rather constant through the canopy rather than increasing progressively from bottom to top of the canopy. We measured the SFPs with our core group of ten lines and found substantial differences in SFPs at the bottom and top of the canopy (Table S3). Top SFP was generally correlated with bottom SFP, as would be expected, but the difference in SPF (bottom–top position) was not correlated with the canopy gradients of protein, oil, or Fe (Fig. 6). Therefore, factors other than the duration of the SFP are responsible for the documented variation in composition with nodal position.

Figure 6 Difference in top/bottom composition traits is not correlated with seed fill period.

The difference in plot normalized composition between the top quad and the bottom quad for protein, oil and iron plotted versus the difference in seed fill period for 51 plots in 2012.

Iron concentrations of soybean seed products

Our results raise the question of whether soy food products made from seed from different portions of the canopy would vary in terms of their mineral concentrations. Three of the most common and simplest products to make from soybean seeds are flour, milk and okara (the particulate material remaining after preparation of milk). Because Fe is one of the most critical minerals to human health and anemia is a global epidemic, we focused our initial analysis on the Fe content of these soy food products. We prepared flour from seven lines, and milk and okara from four lines and Fig. 7 summarizes the results (All elements displayed in File S6). With all three products, the concentration of Fe was highest in products made from seeds produced at the bottom of the canopy and decreased progressively with canopy position of the seeds used. Thus, as would be expected the concentration of seed Fe affects the concentration of Fe in the flour, milk or okara produced from those seeds. Although many questions remain, the public health implications of our findings are apparent. Given that mineral content of seeds, especially Fe, is important our results uncover another source of variation that can be directly exploited.

Figure 7 Canopy differences in iron are reflected in food products.

Fe content of the products from 3 replicates of seven lines (flour) and four lines (Milk and Okara). Boxplots display the five number summary (median, 25, and 75% percentile define the box, with whiskers extending to 1.5 × interquartile range).

The vegetative soybean ionome

The canopy effect on seed mineral concentration prompted us to look at the distribution of minerals in the shoots of vegetative plants. Minerals deposited in seeds are derived from continued uptake from the soil or remobilization of previously accumulated minerals (Hocking & Pate, 1977; Waters & Grusak, 2008), and therefore the leaf ionome of the vegetative plant is relevant to studies of the mature seed ionome. Consequently, we examined the leaf ionome from four genotypes as a function of canopy position. As shown in Fig. 8, the concentrations of Mg, Al, Ca, Mn, Fe, Co, As, and Sr were highest in leaves at the bottom of the canopy and decreased progressively to the top of the canopy. Concentrations of P, S, K, Cu, Zn, Rb, and Mo increased from bottom to top leaves. Na and Ni were present at low absolute concentrations and fluctuated but not in a progressive pattern as for the other minerals. Although leaves at different positions are often analyzed together (or as part of the ‘shoot’), two previous studies with soybean also reported differences in mineral concentrations of lower, middle and upper leaves (Drossopoulos, Bouranis & Bairaktari, 1994) or young and old leaves (corresponding to different node positions) (Duke et al., 2012) that are generally consistent with our results. The basis for differential accumulation of foliar minerals at different positions within the canopy is not clear and will be important to address in future studies. One possible explanation is that the greater phloem mobility of P and K facilitates their enhanced remobilization to upper nodes whereas other less mobile elements (e.g., Fe, Ca, and Mg) tend to remain at their point of initial deposition. This would not readily explain the observed profiles for Cu, Zn and Mo, however, highlighting the complexities involved in metal homeostasis and the significant variation with canopy position. Another working hypothesis could be that K, P, Cu, Zn and Mo are mineral markers of metabolic activity and accumulate in leaves at the top of the canopy that have highest rates of photosynthesis. Because minerals can be remobilized from leaves to developing seeds (Drossopoulos, Bouranis & Bairaktari, 1994; Jiménez et al., 1996; Sankaran & Grusak, 2014), it is tempting to speculate that the canopy seed gradient in Fe and Mg may be related to greater stores of both metals in leaves lower in the canopy. Opposite patterns were observed for other minerals (Ca, Mn, and Cu) suggesting that remobilization is either mineral specific or not quantitatively important in delivery of minerals to developing seeds.

Figure 8 Canopy gradients of leaf composition traits.

For each trait, the data was normalized to the plot average to remove the effect of environment and genotype. The plots display the quadrant average as a line with the 95% confidence interval calculated using standard error as the ribbon. Elements with a significant (p < 1e − 10) effect of gradient in an ANOVA analysis that included Entry, Collection Date and Position.

A final point to note is that the potential exists for some soil particles to adhere to vegetative plant parts, especially lower in the canopy, while seeds are protected from soil contamination by the pods. Since some minerals exhibited opposite patterns, it seems that soil adhesion could not be completely responsible for the patterns observed.

Node position and the developing seed metabolome

Developing seeds were analyzed to determine whether canopy position affected seed metabolism sufficiently to explain the observed differences in protein and oil concentrations at maturity. To do this, we collected developing seeds (cultivar ‘Williams 82’) from the top and bottom of the canopy at several time points over a 24-h period. Because seeds at the top and bottom of the canopy differed in size on the day of the experiment, seeds from the top of the canopy were also collected 6 days later when they had reached the same size (fresh weight seed−1) as the bottom seeds on the first collection date. All seeds were at the stage of development where cell expansion and accumulation of storage compounds (protein and oil) were the dominant metabolic processes (Collakova et al., 2013). Untargeted metabolite profiling was conducted for analysis of polar compounds, free amino acids, free fatty acids, and total fatty acids (File S7).

In general, most metabolites did not show diurnal changes in concentration, but there were differences in concentrations as a function of seed size and node position. The metabolite plots in Fig. 9 illustrate some of the different patterns observed. The concentration of sucrose (Fig. 9A) in developing seeds did not vary diurnally and remained relatively constant but the concentration was slightly higher in the smallest seeds (day 1, top seed) compared to the larger seeds sampled at the bottom position on day 1 or top position on day 7. The decrease in sucrose concentration comparing top seed on day 1 and day 7 likely reflects in part the dilution effect caused by storage product accumulation as the seeds increased in size by roughly 2-fold. In contrast, the concentration of citrate in developing seeds was roughly equal among the three samples (Fig. 9B). These results suggest that seeds actually accumulate sucrose and to a larger extent citrate as they increase in dry matter during seed fill (thereby negating the dilution effect caused by seed growth). This also indicates that developing seeds have ample sugars and organic acids irrespective of size and node position and time of day. In marked contrast to sucrose and citrate were the dramatic differences observed in free asparagine (Asn) concentration (Fig. 9C), which was highest in top seed sampled on day 1 (Aug 20), and lowest in bottom seed sampled on the same day; the difference was roughly 8-fold. Sampling top seed on day 7 (Aug 26), when seed size was equivalent to that of bottom seed on day 1, still resulted in a ∼4-fold elevation of free Asn concentration. The roughly 2-fold decrease in Asn concentration in seeds at the top of the canopy from day 1 to day 7 likely reflects the dilution effect of growth. The pattern for Asn concentration is potentially of interest because free Asn concentration during seed development correlates with protein concentration at maturity (Herman, 2014; Hernandez-Sebastia et al., 2005; Miller et al., 2008; Pandurangan et al., 2012). The results obtained in the present study suggest that greater supply of Asn to developing seeds at the top of the canopy may contribute to the observed greater accumulation of storage protein.

Figure 9 Concentrations of selected primary metabolites in developing seeds of cultivar ‘Williams 82’.

A, Suc; B, citrate; and C, Asn. Boxplots display the five number summary (median, 25, and 75% percentile define the box, with whiskers extending to 1.5 × interquartile range) for three replicates at each sampling time: 7 AM (7) , 12N (12) , 7 PM (19) and the following morning at 7 AM (31). The black vertical bars represent the intervening night period. Values are µg (g DW).

Importantly, Asn was also one of the important metabolites that distinguished the three sets of seeds collected based on a global metabolite analysis (File S8). Mean values for Asn, and other protein amino acids are shown in Fig. 10. The concentrations of the free amino acids was highest in the small seed (top seed, day 1). Concentrations of Ala, Asn, Gly, and Thr were substantially higher in top seed at day 7 relative to bottom seed at day 1 (when seed sizes were similar). Of those amino acids, Asn was present at the highest absolute concentrations and may contribute to the storage protein biosynthesis either by acting as a signal metabolite or providing substrate for protein biosynthesis.

Figure 10 Concentrations of free amino acids in developing seeds.

Boxplots display the five number summary (median, 25, and 75% percentile define the box, with whiskers extending to 1.5 × interquartile range) for values from each sampling interval (3 replicates and 4 time points are merged within each box) and nodal position. Ornithine levels reflect both ornithine and arginine as arginine is converted to ornithine during sample prep for GC-MS. D1.bot, D1.top and D7.top refer to the samples collected on day one top and bottom quadrants and the day seven top quadrant respectively.

Discussion

The present study yields two major conclusions. First, the position along the main stem at which soybean seeds develop has a profound impact on seed composition, affecting the concentrations of protein, oil and certain minerals at maturity. Second, the canopy position effects on seed mineral concentrations (in particular Fe) are sufficiently large that there may be direct implications for human nutrition in countries where plants are the main source of protein and soybeans are used for human food. These conclusions are discussed in more detail below.

Positional effects on seed protein and oil concentration are broadly observed

Results of the present study demonstrate that for 10 lines grown over a period of 3 years there were remarkably consistent gradients in protein and oil concentrations in mature seeds as a function of nodal position (Figs. 2A and 3). Increased concentration of oil in seeds from lower nodes could result from the increased duration of the SFP documented for lower pods (Table S3) because the accumulation of oil in seeds often starts earlier than protein (Rotundo & Westgate, 2009; Saldivar et al., 2011). However, oil accumulation tends to plateau before protein accumulation and therefore, percent oil will often decrease with increasing duration of the SFP rather than increase. In the present study, the protein and oil concentration gradients from bottom to top of the canopy were not correlated with the difference in SFP between the two positions (Fig. 6) and thus it appears that SFP does not determine the observed gradients in protein and oil concentration. Micro-environment appears to be one factor controlling protein and oil concentration gradients in the canopy because removal of neighboring plants at flowering increased protein concentration at all positions and decreased the difference between top and bottom nodes (Fig. 5). While it is not clear which micro-environmental factor(s) might actually be involved, we suggest that increased light energy reaching lower leaves may be a contributing factor. Metabolomic analysis of developing seeds that identified free Asn as one of the primary metabolites distinguishing seeds at the bottom and top of the canopy supports this conclusion. Asparagine is the major free amino acid in developing soybean seeds and differences in Asn concentration during development are positively correlated with protein concentration at seed maturity (Hernandez-Sebastia et al., 2005; Pandurangan et al., 2012). Furthermore, over-expression of asparaginase in soybean, driven by an embryo-specific promoter, resulted in a reduction in free Asn concentration during development and reduced protein concentration in mature seed, measured by nitrogen analysis (Pandurangan et al., 2015). Collectively, these results suggest that free Asn is a sensor or regulator of processes that determine protein accumulation in soybean seeds (Herman, 2014). Our results are consistent with this hypothesis and suggest that differences in free Asn concentration may explain the position effects on seed protein (and oil) concentration. Nitrogen and carbon flux into pods is largely provided by nearest sources (Seddigh & Jolliff, 1986; Streeter & Jeffers, 1979) including the nearest trifoliolate leaves. We speculate that decreased light at lower positions in the closed canopy (i.e., with neighboring plants) would reduce leaf metabolism as well as the xylem flux of ureides and/or nitrate from roots to the lower leaves, thereby restricting the ability of those leaves to provide Asn (and Gln) to developing seeds. In contrast, removal of neighboring plants (in the ‘thinned’ plant treatment) would increase light at lower nodes thereby enhancing overall leaf metabolism and the flux of reduced nitrogen to subtending pods resulting in increased protein (and reduced oil) accumulation.

Positional effects on seed mineral concentration are documented

The concentration of minerals in seeds reflects the combined action of transport processes and regulation at multiple steps starting with mobilization from the soil, uptake into the root, and transport to the shoot for distribution among organs (Grusak, Dellapenna & Welch, 1999; Waters & Grusak, 2008). Deposition of some minerals in seeds can also involve remobilization from leaves during seed filling (Grusak, Dellapenna & Welch, 1999; Hocking & Pate, 1977), and it is interesting that different minerals show fundamentally different profiles of accumulation in seeds as a function of canopy position (Fig. 3). These differences could reflect alternate routes from the apoplast to the symplast or differences in mobility in the phloem (White, 2012). Interestingly, minerals that tended to have highest concentrations in seeds at the bottom of the canopy (e.g., Mg, Fe, and Cu) are considered to have moderate to good phloem mobility compared to the minerals that tended to concentrate in the top of the canopy such as Mn (and in some cases Ca) that are considered to have poor phloem mobility. These results suggest that remobilization from leaves may be playing some role at least in the positional effects on the mature seed ionome. Another factor that may impact the distribution of minerals in seeds along the mainstem is precipitation. This speculation is based on the increased concentrations of Ca, Mn, and Sr found in seeds at the top of the canopy in 2010, which had above normal precipitation. It is possible that increased precipitation resulted in greater xylem transport of certain minerals (including Ca, Mn, and Sr) to developing seeds at the top of the canopy, or alternatively, that weather conditions in 2010 allowed greater remobilization of selected minerals from leaves via the xylem. It is recognized that while Ca and Mn are generally considered to have very low phloem mobility and are therefore not remobilized from senescing leaves, there is variation among species in the extent of remobilization (Maillard et al., 2015). Conceivably, remobilization may also be triggered from leaves of all species under certain conditions.

While multiple seed constituents exhibited canopy concentration gradients, it seems unlikely that they are all caused by the same factors. Changing the microenvironment by thinning plants to allow increased light penetration into the canopy altered the protein and oil gradients but did not affect observed gradients for most of the minerals (Fig. 5). Furthermore, while the slope of many gradients changes across lines, treatment and year, the way that they change is not well correlated between the different constituents, as illustrated in the plot normalized correlation matrix (Fig. 4A), where relatively few strong correlations among the various parameters were apparent. However, numerous correlations were apparent when mean plot values were compared (Fig. 4B). Several minerals (e.g., P, Mn, Fe, Zn, S, and Co) had a negative relationship with oil concentration and increased with protein concentration. Thus, some coordination between seed storage product accumulation and mineral uptake into seeds is evident. However, the results suggest that total uptake of a mineral and the allocation among nodal positions are controlled by different mechanisms, and in general, canopy positional effects on minerals and protein/oil appear to be controlled by distinct mechanisms. It should be noted that altering the microenvironment by thinning plants did affect the observed gradients in seed concentrations of Ca, Mn, and Sr, which were also the minerals altered in distribution in 2010 (the year of this study with above normal precipitation). These results highlight the differences among minerals in terms of factors controlling their distribution among seed produced at different node positions. Clear, continued studies in the future will be required to sort out the different mechanisms involved.

Human nutrition implications for variation in seed composition

Soybeans are valued for their protein and oil content, but when used for human nutrition the content of minerals such as iron and zinc is also critically important. On a global scale, human iron deficiency is one of the most prevalent nutritional disorders (McLean et al., 2009) especially in countries where plant-based diets are prominent. As discussed above, nodal position affected the concentration of several minerals such as Mg, Fe, and Cu that were present at higher concentrations in seeds produced at the bottom of the canopy. Iron is of particular interest and was generally 20% higher in seeds produced lower in the canopy relative to the top and as expected, differences in seed iron concentrations affected the concentration of iron in soy food products made from those seeds (Fig. 7). Soy flour preserved more Fe than did milk; perhaps mineral retention improvement through product preparation is possible. An immediate application of our results with respect to human nutrition would be to use seeds from the top and bottom halves of the canopy for different purposes, with seeds produced in the lower half reserved for production of iron-rich soy foods for human consumption. Thus, knowledge of these canopy position effects provides an unexpected approach to link agronomic practices to improve human nutrition and health.

New type of seed heteromorphism and implications for climate change impacts

Seed heteromorphism is well established (Matilla, Gallardo & Puga-Hermida, 2005) but the seed heterogeneity documented here establishes a new category where an individual plant produces a continuum of seeds that differ in major aspects of their composition (protein, oil, and minerals) but are morphologically very similar. Overall, our results raise a number of questions and directions for future research. For example, it would be interesting to explore whether there are positional effects on soybean seed functional traits such as seed vigor or seedling stress tolerance. Because environment during reproductive development of plants is now recognized to broadly impact seed properties, such as growth performance and stress tolerance of the progeny (Biodner et al., 2007; Tricker et al., 2013), it will be interesting to further explore similar properties of soybean seed produced at the different parts of the canopy. Our results also raise the question of whether similar effects occur in other species including non-domesticated plants where there might be some ecological significance.

Another area that will be interesting to explore is the impact of elevated CO2 on the canopy positional effects described in the present study. It was recently reported (Loladze, 2014; Myers et al., 2014) that grain from many species, including soybean, have lower concentrations of Zn and Fe when plants are grown at elevated CO2 thereby uncovering a new climate change challenge to global health. The meta-analysis established a ∼5% reduction in soybean seed Fe and Zn concentrations at high CO2. It is relevant to note that variation in seed Fe concentration with node position established in the present study is substantially larger (4-fold greater) compared to the impact of climate change on mean seed Fe concentration. Therefore, our results are likely to be meaningful from a quantitative standpoint and have important implications for examining the impact of climate change on the seed ionome. For example, it will be interesting to determine how this overall reduction in mean seed Fe concentration at elevated CO2 is related (if at all) to canopy position effects; is Fe reduced 5% in seeds at nodes throughout the canopy or are certain positions affected to a greater degree than others? Identifying the molecular mechanisms underlying canopy gradients in composition may provide new approaches to controlling soybean seed quality for various uses, including food for human consumption under conditions of global climate change.

Supplemental Information

File S1 Raw Gradient plots for each line/year combination

Values are PPM (Elements), mg (SampleWeight), and Percentage (Protein/Oil).

Click here for additional data file.

File S2 All Compositional Traits Normalized Gradients

Click here for additional data file.

File S3 Interactive Correlation plots of all plot normalized data

Click here for additional data file.

File S4 Interactive Correlation plots of plot mean data

Click here for additional data file.

File S5 Interactive Correlation plots of all data, not normalized

Click here for additional data file.

File S6 Composiition of food products for all elements

Click here for additional data file.

File S7 Metabolomics analysis scheme

Click here for additional data file.

File S8 Analysis of metabolome of developing soybean seeds

(A) PLS-DA scores plot (R2 = 98.7%, Q2 = 81.1%, P < 0.001 by permutation test) of soybean seeds at different canopy position and time of day. (B) Variable Importance in the Projection (VIP) for the first component showing the fifteen most important compounds.

Click here for additional data file.

Table S1 Indeterminate lines used in the present study and selected characteristics

Click here for additional data file.

Table S2 Weather summary (June 1–August 31) during the 2010 to 2012 growing seasons

Click here for additional data file.

Table S3 Genotype differences in Seed fill period (SFP) and the difference in SFP at two node positions (bottom minus top; delSFP)

Click here for additional data file.

The authors thank Kunming University of Science and Technology (KUST), Kunming City, P.R. China for supporting the visit of Prof. Kunzhi Li to UIUC; Karl E. Weingarter and Marilyn L. Nash for advice on soy food preparation and providing access to the Test Kitchen facility of the National Soybean Research Laboratory at the University of Illinois and Greg Ziegler for expert technical assistance with ionomic analysis.

Additional Information and Declarations

Competing Interests

Author Contributions

Data Availability

The authors declare there are no competing interests.

Steven C. Huber and Ivan Baxter conceived and designed the experiments, performed the experiments, analyzed the data, wrote the paper, prepared figures and/or tables, reviewed drafts of the paper.

Kunzhi Li and Catherine M. DeMuro performed the experiments, reviewed drafts of the paper.

Randall Nelson conceived and designed the experiments, performed the experiments, contributed reagents/materials/analysis tools, reviewed drafts of the paper.

Alexander Ulanov performed the experiments, analyzed the data, reviewed drafts of the paper.

The following information was supplied regarding data availability:

The raw data can be downloaded from here: http://www.ionomicshub.org/home/PiiMS/fileDownload?file=50.

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
