# Peer review of "Canopy position has a profound effect on soybean seed composition"

_PeerJ, doi:10.7717/peerj.2452_

## Round 0.1 · original submission · Minor Revisions

Overall the reviews were quite positive. There are some suggestions for improvement by two reviewers, which should be incorporated in the revised manuscript.

·

Basic reporting

This article brings significant contribution to the understanding of the physiological mechanisms underlying the effect of canopy position on soybean seed constituents. It is written in good English presenting sufficient introduction and background regarding the broad field of knowledge. Figures are relevant to the content of the article with good description and labelling. Resolution of figure 7 and supplemental figure 1 is of poor quality. Please, improve these two figures resolution. All results are relevant to the hypothesis, even though it could still be improved. I would suggest Fe bioavailability experiments for the food products and expression analysis of key genes involved in metal homeostasis during seed filling.

Experimental design

The article meets all standards for publication.

Validity of the findings

The article meets all standards for publication.

Additional comments

The article is composed of simple but well-designed experiments. Briefly, the canopy gradient of seed composition of ten soybean lines was measured in a period of 3 years time. Raw data was properly normalised to avoid fluctuations related to the environment and genotypes. The authors were able to show that seed protein, oil and minerals accumulate in different patterns according to the canopy gradient. A simple and elegant experiment showed that protein and oil accumulation is regulated by a process other than that of mineral distribution in soybean seeds.

The vegetative soybean ionome could have been linked to the expression analysis of key genes involved in metal homeostasis and remobilisation. However, results presented are satisfactory to support the main idea of this work. Results regarding the Fe concentration of food products made of soybean seeds coming from different canopy positions are interesting and of great importance for the field of human nutrition and health in developing countries. Also, the main novelty of this work is the new insights about the physiological mechanisms that may underlie these positional effects.

Having said that I strongly recommend this paper for publication.

Reviewer 2 ·

Basic reporting

The manuscript show interesting results comparing oil, protein and mineral concentrations of soybean seeds seeds in different positions in the canopy. The results reproduce and confirm oil and protein inverse concentration gradients from bottom to top, which was known before. Authors also show that some elements are more concentrated in seeds from the bottom of the pods. Especially, Fe is among these, and authors demonstrate that soybean processed products can be enriched in Fe if prepared from bottom-derived seeds, a simple and yet important application of the findings. This study demonstrate that morphologically similar seeds have very different ionome profile, and correlate some of these changes with protein and oil gradients. The manuscript is of great quality and should be accepted for publication.

I made a few specific suggestions below that could improve the data presentation (with inclusion of some supplementary data) and text:

• Line 130: this statement could be highlighted in the abstract.
• Figure 3: would be better to have figures presented in the same order as they are cited in text.
• Line 146: would be nice to include the K data, so readers can compare with Rb.
• Line 182-184: the correlation between protein, Zn and Fe was found in seeds of other plants as well. Could cite the work by Uauy et al (2006, Science, 314, 1298) that provides some mechanistic insight about it.
• Line 198: Phrase “There was one also a strong negative correlation” should be corrected.
• Line 239: I assume that authors have the results for other elements as well. Do the elements that have higher concentration at the bottom also result in higher concentration in the products derived from these seeds? Would be interesting to have the results as a supplementary figure.
• Fig 9: A, B and C parts of the figure are not well pointed in the image.
• Line 445: I’d suggest to mention the potential of the results to interact with climate change-derived alterations in Fe/Zn concentration in the abstract.

Experimental design

No comments

Validity of the findings

Please see comments above.

·

Basic reporting

Clearly written manuscript, with well defined objectives in relation with a correct analysis of literature. Figures are well designed and relevant but the labeling could be improved (see suggestions).

Experimental design

All methods used are well described, experimental procedures and analysis are relevant. Results obtained revealed the complexity of seed filling when considering all required nutrients, and the role of leaf for remobilization, according to the canopy structure.

Validity of the findings

The manuscript by Huber and collaborators provides a wealth of data on the effect of canopy position on protein and oil concentration, seed and leaf ionome in soybean (10 soybean lines, 3 years). The duration of the seed filling period as well as the impact of microenvironment on seed ionome are considered. It also shows that the composition of soy food products is directly linked to the seed ionome. The link between the leaf soybean ionome and the seed ionome has also been investigated. It is to my knowledge the first time that such an impressive amount of data has been collected to evaluate the impact of canopy position on seed ionome.

Additional comments

General comments:
The manuscript by Huber and collaborators provides a wealth of data on the effect of canopy position on protein and oil concentration, seed and leaf ionome in soybean (10 soybean lines, 3 years). The duration of the seed filling period as well as the impact of microenvironment on seed ionome are considered. It also shows that the composition of soy food products is directly linked to the seed ionome. The link between the leaf soybean ionome and the seed ionome has also been investigated. It is to my knowledge the first time that such an impressive amount of data has been collected to evaluate the impact of canopy position on seed ionome.

1. The Introduction lists two factors affecting the development of seed at the top of the plant differently than those at the bottom of the canopy: the seed filling period, and the role of canopy microenvironment. Yet the impact of remobilization or rather the impact of nutrient mobility is discussed further. This aspect could be added in the introduction.

2. I can appreciate the data here and the amount of effort that goes into a study like this one. The data looks quite interesting and will be informative for readers. However, the figures could be improved to reach more homogeneity that will help the readers. For example, the comparison between raw data provided in Figure 2 and normalized data provided in Figure 3 would be facilitated by the establishment of a precise order of seed composition traits: always “SampleWeight” in first, “Percentage of Proteins” in second, etc… In the same way, the different elements of ionome never appear in the same order in the different figures. It will help the reader to identify the data when focusing on only one element..

Figure 2 in which the raw data is supplied is very interesting but this figure and the following need more information on the figure itself and in the legend. For example, what is the unit of “SampleWeight” (individual seed weight ? as explained page 5 Line 120). It should be added in the legend of each figure, and reminded for proteins, Oil, and Fe (Figure 2). Please place the label “A” and ”B” in figure 2.

What is the real meaning in the different legends of “(p<1e-10)” ?

Figure 9 : Unit of the Y axis for sucrose is probably wrong. Which line(s) was used for these analysis ? This should be indicated in the legend

3. Line 282, the statement “Opposite patterns were observed for other minerals (Ca, Mn, K and Cu) suggesting that remobilization is either mineral specific or not quantitatively important in delivery of minerals to developing seeds.” was made. We have an opposite pattern for Ca, Mn and Cu. However, there is no obvious canopy gradient of seed for K, only in leaves. As a result, readers may not conclude to an opposite pattern for K.
Similarly, line 262, authors state that Se were highest in leaves at the bottom of the canopy as shown in Fig. 8 but Se quantification is not given in the figure.

4. Line 169, a highly correlated element pair as illustrated in the normalized plot correlation matrix (Figure 4A) has been forgotten: Ca- Mn.

5. As the authors noted, differences of nutrient accumulation in seeds as a function of canopy position could reflect the differences of mobility in the phloem. It has already been shown that given minerals tended to have a high mobility in phloem while others have a low mobility in phloem. Then references need to be cited, lines 390 (White, 2012 for exeample). The particular effect in 2010 on Ca, Mn and Sr could be explained in the discussion (line 392). Indeed, the weather conditions and more particularly the above normal precipitations could be responsible of these modifications. The water availability is then probably linked to the ability of Ca and Mn to be remobilized via the xylem (Maillard et al., 2015 ; Dayod et al., 2010 ; Malone et al., 2002 ; Nable et Loneragan, 1984….). Theses paper could be used to broaden the discussion on leaf remobilization (lines 281-283).

In the same way, in the discussion (lines 394-395), the authors don’t return on the impact of microenvironment on the concentrations of minerals. Only the sentence “Changing the microenvironment altered the protein and oil gradients but did not affect observed gradients for most of the minerals” gives the conclusion. Yet, the microenvironment has an impact on Ca, Mn and Sr, the same elements affected by particular conditions of 2010. This point would merit further discussion.

Minor comments :

Page 1, line 4 : what the aim of the second symbol after DeMuro ?
Line 134 : not really obvious for Zn, except in 2010, or Sr (line 140)
Line 199 : There is also a strong correlation between S and Mo that could be noted
Line 262, Se in not shown in Figure 8
Line 362 : does not ?
Line 502: mg/ml, line 504: mg/mL, line 530: µgml-1 – please use uniform units.

---

## Round 0.2 · accepted · Accept

Congratulations on the acceptance of your manuscript.

Reviewer 2 ·

Basic reporting

The article is accepted.

Experimental design

The article is accepted.

Validity of the findings

The article is accepted.

Additional comments

The article is accepted.

·

Basic reporting

See comments of the first submission

Experimental design

See comments of the first submission

Validity of the findings

See comments of the first submission

Additional comments

The new version of the manuscript has been improved enough to be suitable for publication in Peer Journal